# Exosome-Laden Scaffolds for Treatment of Post-Traumatic Cartilage Injury and Osteoarthritis of the Knee: A Systematic Review

**DOI:** 10.3390/ijms242015178

**Published:** 2023-10-14

**Authors:** Jorden Xavier, William Jerome, Kenneth Zaslav, Daniel Grande

**Affiliations:** 1Albert Einstein College of Medicine, New York, NY 10461, USA; jorden.xavier@einsteinmed.edu (J.X.); william.jerome@einsteinmed.edu (W.J.); 2Feinstein Institute for Medical Research, New York, NY 11030, USA; kzaslav@northwell.edu; 3Department of Orthopedic Surgery, Lenox Hill Hospital, New York, NY 10075, USA; 4Department of Orthopedic Surgery, Long Island Jewish Medical Center, New York, NY 11040, USA

**Keywords:** exosomes, scaffold, osteoarthritis, knee, articular cartilage, trauma

## Abstract

Mesenchymal stem cell (MSC)-based exosomes have garnered attention as a viable therapeutic for post-traumatic cartilage injury and osteoarthritis of the knee; however, efforts for application have been limited due to issues with variable dosing and rapid clearance in vivo. Scaffolds laden with MSC-based exosomes have recently been investigated as a solution to these issues. Here, we review in vivo studies and highlight key strengths and potential clinical uses of exosome–scaffold therapeutics for treatment of post-traumatic cartilage injury and osteoarthritis. In vivo animal studies were gathered using keywords related to the topic, revealing 466 studies after removal of duplicate papers. Inclusion and exclusion criteria were applied for abstract screening and full-text review. Thirteen relevant studies were identified for analysis and extraction. Three predominant scaffold subtypes were identified: hydrogels, acellular extracellular matrices, and hyaluronic acid. Each scaffold–exosome design showcased unique properties with relation to gross findings, tissue histology, biomechanics, and gene expression. All designs demonstrated a reduction in inflammation and induction of tissue regeneration. The results of our review show that current exosome–scaffold therapeutics demonstrate the capability to halt and even reverse the course of post-traumatic cartilage injury and osteoarthritis. While this treatment modality shows incredible promise, future research should aim to characterize long-term biocompatibility and optimize scaffold designs for human treatment.

## 1. Introduction

Osteoarthritis (OA) of the knee is a chronic debilitating condition that affects over 30% of the world’s population over the age of 45 [1]. While normally a degenerative disease, 25–50% of OA patients report post-traumatic osteoarthritis of the knee as a result of trauma-induced chondral or osteochondral defects, as well as meniscal and/or ligamentous injuries [2,3]. Current conservative treatment options include non-steroidal anti-inflammatory drugs (NSAIDs), physical therapy, and intraarticular glucocorticoid injections. These methods provide only temporary improvements in pain and mobility and fail to halt or reverse pathological progression to end-stage disease. As a result, total knee arthroplasty (TKA) remains the gold standard treatment option for knee OA [1]. While viable for patients over the age of 70, TKA is contraindicated for younger patients due to higher physical demands that lead to subsequent implant failure [4,5]. Given that the median age of onset for post-traumatic osteoarthritis is 55, there is a pressing need to develop novel non-surgical therapeutics that can halt disease progression in younger adults [5]. 

One feasible option that has been explored as a therapeutic treatment for post-traumatic cartilage injury of the knee is the use of mesenchymal stem cells (MSCs). MSCs have been shown to reduce inflammation, stimulate cellular differentiation and proliferation, and regulate metabolic pathways in ways that slow pathologic progression [6,7]. The therapeutic effects of MSCs in treating cartilage injuries stem from the paracrine release of extracellular vesicles (EVs). EVs are nanoparticles that contain proteins, lipids, mRNA, and microRNA used for intercellular communication [8]. The two major subclasses of EVs include exosomes and microvesicles, with exosomes serving as the primary effector molecules given their bilayer membrane which enables long distance travel through tissue as well as protection of genetic material [9]. In both in vitro and in vivo studies, exosomes have shown great promise in alleviating and even reversing the effects of knee OA. He et al. demonstrated that exosomes inhibited pro-inflammatory markers such as IL-1B while upregulating collagen-promoting genes such as COL2A and ACAN in rat chondrocytes [10]. Furthermore, Tao et al. showed that rat models treated with exosomes displayed reduced joint wear and cartilage matrix loss in comparison to control groups [11]. 

The progress of developing MSC–exosome-based therapies, however, has been limited since injected exosomes are cleared from the repair site within hours [12]. Furthermore, harvesting consistently large quantities of exosomes has been limited due to large degrees of variability in the methodology [13]. To address these issues, bioengineered scaffolds have recently been considered as a possible vehicle for exosome delivery. The current literature has extensively reviewed the progress made in understanding the influence of free exosomes in treating cartilage injuries and knee OA. However, no efforts have been made to examine the promising results of early investigations into exosome therapies using scaffolds as drug delivery vehicles. Here, we review the advances made in exosome–scaffold therapies in the treatment of post-traumatic cartilage injuries and subsequent osteoarthritis of the knee.

## 2. Results

A total of 466 studies were obtained from the initial query search after removal of duplicate papers. After abstract screening and full-text review, 12 studies met the defined inclusion and exclusion criteria for analysis. Table 1 summarizes the key characteristics and experimental design of each exosome–scaffold therapy investigation. 

Six of the twelve studies utilized human- or animal-derived bone marrow MSCs as an exosome source. Other common exosome sources included MSCs derived from immortalized pluripotent stem cell lines and umbilical cord products. Tao et al. were the only team to report the utilization of MSCs derived from the synovial membrane and/or fluid [21]. Dosage of exosomes in the scaffolds ranged from 44 to 3000 μg/mL or 10^9^ to 10^11^ particles/EV, as measured by NTA. Three studies reported multiple exosome injections throughout the study; these injections were either a dosage of exosome and scaffold or exosomes only into an already-implanted scaffold. All studies adhered to MISEV guidelines and reported exosome isolation techniques and characterization via NTA, electron microscopy, and/or Western blotting [26].

The most commonly used animal models were rats and rabbits, with two studies utilizing mice and micropig models, respectively. To induce traumatic cartilage injury, 11 of the 12 studies created drilled cylindrical osteochondral or chondral defects in the femoral condyle or articular cartilage, respectively. Two studies created traumatic osteoarthritis models via ligament transection: Pang et al. transected the medial meniscus; while Tao et al. removed the medial meniscus along with the anterior cruciate and medial collateral ligaments [19,21]. 

Table 2 summarizes the therapeutic effects of the exosome–scaffold models utilized in the 13 studies. Reported gross findings across all 12 studies were similar overall: usage of exosome–scaffold therapies led to the generation of smooth cartilage that filled the defect and was contiguous with surrounding native tissue. Furthermore, reported histological findings were also similar across all 12 studies, with exosome–scaffold models leading to increased chondrocyte proliferation and hyaline cartilage formation. International Cartilage Repair Society (ICRS) scores were utilized by 10 of the 12 studies to quantify the macroscopic and histologic changes seen in each experimental group. Five studies reported quantitative grading scores, while six studies used this scale to qualify differences between exosome–scaffold treatments and controls. Biomechanical testing was performed in 3 of the 12 studies to evaluate the performance of cartilage generated by the exosome–scaffold model. Lastly, gene expression within the site of treatment was analyzed in 11 of the 12 studies to qualify the influence of exosomes in regulating pro- and anti-inflammatory processes.

### 2.1. Hydrogels

Hydrogels were the most commonly used scaffold type, as seen in 7 of the 12 reviewed studies. These scaffolds were pre-seeded with exosomes and surgically implanted into established cartilage defects within the animal models. Macroscopic ICRS scores for hydrogel-based scaffolds ranged from 10.7 to 11.25, while histologic ICRS scores ranged from 11.2 to 16.1. Both scores were obtained at 12 weeks post-implant. The exosome release profiles in hydrogel scaffolds varied widely, ranging from 7 to 35 days. All reported exosome–hydrogel models displayed upregulation of COL2 and downregulation of COL1, which is consistent with healing as opposed to scarring. Many of these systems also showed a decrease in pro-inflammatory gene markers and an increase in M2/M1 macrophage ratio, which is also consistent with tissue healing and recovery.

### 2.2. Acellular Extracelllular Matrices

Two studies utilized an acellular extracellular matrix (AECM) derived from porcine knee joint articular cartilage. These scaffolds were surgically implanted into established cartilage defects within the animal models, after which they were seeded with exosomes in situ via multiple injections over time. Both studies reported higher ICRS scores in AECM exosome models when compared to controls. Yan et al. reported small amounts of fibrous tissue with increased chondrocytes at 4 weeks post-implant that improved at 8 weeks [23]. Jiang et al. observed a gap between repaired tissue and surrounding normal cartilage at 3 months post-implant [16]. However, at 6 months, complete fusion of scaffold and normal cartilage was observed. In regard to gene expression, Yan et al. reported an increase in COL2 and proteoglycans, while Jiang et al. reported an increase in M2 macrophages and IL-10, both of which are anti-inflammatory indicators [16,23]. 

### 2.3. Hyaluronic Acid

Hyaluronic acid (HA) scaffold models were analyzed in 3 of the 12 studies. These scaffolds were pre-seeded with exosomes and injected into established cartilage defects within the animal models. Zhang et al., in 2022, reported an ICRS macroscopic score of 9.22 at 4 months post-implant, while Wong et al. reported a score of 10.33 at 12 weeks post-implant [22,25]. Liu et al., in 2017, reported that their exosome-HA model outperformed all other experimental groups by having the highest ICRS scores; however, the exact values were not reported [17]. With regard to the biomechanical performance of newly generated cartilage, Wong et al. reported a Young’s modulus of 13.41 kPa and a stiffness of 7.02 N/mm at 12 weeks post-implant [22]. Furthermore, Zhang et al., in 2022, reported a Young’s modulus of 15.14 kPa and a stiffness of 7.94 N/mm [25]. Liu et al. reported upregulation of COL2 and downregulation of COL1 [17]. Wong et al. confirmed these findings while also reporting greater areal deposition of GAG [22].

### 2.4. Risk of Bias

Table 3 outlines the results of our risk of bias assessment undertaken to determine the quality of each study. Examined biases included selection bias, performance bias, detection bias, attrition bias, reporting bias, and other sources of bias. Nine of the twelve studies reported random allocation of animal models within the experimental groups and were therefore assigned a low risk for selection bias in terms of random sequence allocation. The methodology of randomization, however, was not specified in any of these studies. The other three studies were assigned an unclear risk status. Only five studies reported blinding of personnel when assessing outcomes and were therefore classified as having low risk of blinding with regard to detection bias; the other seven studies were classified as having unclear risk. All studies reported some form of quantitative results that were presumably averaged out across all animal samples regardless of experimental group, thereby rendering these studies low risk for detection bias in terms of random outcome assessment and for reporting bias. All studies were assigned low risk for attrition bias status since none of the animals were reported to have died prior to the conclusion of the studies. None of the studies outlined any methods for allocation concealment; as a result, they were assigned unclear risk status for this aspect of selection bias. 

## 3. Discussion

Traumatic injury to the knee has been shown to contribute significantly to the development of knee OA, especially in younger patients. Joint injuries such as osteochondral defects, ACL or meniscus tears, fractures, and joint dislocations lead to extensive damage to the surrounding articular cartilage, which, in turn, predisposes the cartilage to inflammation and subsequent deterioration [27,28,29,30,31]. While treatments exist to provide symptomatic relief and improve joint mobility, none so far have been effective in halting or reversing disease progression. Exosomes have garnered popularity in recent years due to their therapeutic benefits in treating OA of the knee. Numerous studies using chondrocyte and animal models have documented the direct biological effects that exosomes exert to inhibit pro-inflammatory signaling and upregulate articular cartilage and subchondral bone regeneration [32,33,34]. Postulated mechanisms of exosome action include stimulation of macrophage cytokine/chemokine production, as well as direct transfer of genetic material into chondrocytes to induce collagen formation and attenuate protein and extracellular matrix degradation [35]. Furthermore, clinical applications of exosome-containing MSCs have shown positive results, with patients reporting reduced pain, stiffness, and immobility [36].

Despite these promising results, advancements in exosome therapeutics have been hindered by two critical issues: unreliable exosome collection methods and a lack of sustained delivery in vivo. Exosome isolation from biological fluids often leads to co-contamination with other particles such as chylomicrons and lipoproteins due to their size overlap (30–150 nm) [37,38]. While numerous techniques and devices have been developed to improve exosome harvest, there is a high degree of variability in terms of final yield, purity, and quality [39,40,41]. Once exosomes are injected into the body, they are rapidly cleared from blood circulation and accumulate in the liver, lung, spleen, and GI tract, regardless of the method of injection [42,43]. Studies have shown that isolated exosomes have a reduced half-life and are quickly taken up by reticuloendothelial macrophages to be removed from circulation [44,45].

To address these two major hurdles, engineered scaffolds have been investigated as a potential drug delivery vehicle for exosomes to treat post-traumatic cartilage injury and osteoarthritis of the knee. The use of three-dimensional scaffolds creates an environment that maintains expression of MSC-based exosome surface markers and promotes cell-to-cell interactions via paracrine signaling. An in vitro study by Rocha et al. in 2018 showed that three-dimensional scaffold architecture favored an upregulation of certain microRNA lines, thereby enhancing the cellular effects of exosomes within the culture environment [46]. Furthermore, scaffolds designed to mimic the biomechanical behavior of native articular cartilage tissue can provide biophysical cues to further influence exosome stimulation of both chondrocytes and the extracellular matrix [21]. Lastly, three-dimensional scaffolds create an environment conducive to controlled exosome release via hydrophilic and hydrophobic interactions that alter with scaffold degradation and/or integration into surrounding tissue [13]. Given the potential benefits that these biologic solutions can provide for patients, this systematic review aimed to collate different exosome–scaffold therapies and compare their abilities across multiple treatment facets to halt or reverse post-traumatic knee OA and cartilage injury.

The studies in our review detailed a variety of scaffold types and compositions, with each displaying a different release pattern for the exosomes they carry. Exosome release ranged anywhere from 5 days to 35 days, with 80–100% of pre-loaded exosomes being taken up within the knee joint. This variability may allow for future customization of scaffolds based on the needs of an individual that suits their lifestyle. While these customizations may vary drastically, in most reported cases, a desired effect can be seen as soon as 4–6 weeks post operation if delivery is optimized. Some groups allowed this healing process to continue over a longer time frame without subsequent treatment and saw further improvement. This suggests that the changes made at the early parts of the healing process can lead to effects that continue to aid recovery weeks to months later. This idea was also posited by Wong et al. and Zhang et al. in 2021 [22,24]. With regard to the release profile of these exosomes, it is important to note that each study used a different method of quantifying these patterns, many of which were in vitro and may not represent release in vivo. 

Exosomes are extracellular vesicles that contain macromolecules enabling cell-to-cell communication, but the physiological purpose of their generation is largely unknown. Exosomes vary from cell type to cell type; however, MSC-based exosomes have been of particular interest due to their anti-inflammatory effects [47]. In this review, we examined exosomes derived from several sources including bone marrow MSC, umbilical cord MSC, and immortalized clonal MSC. While there is expected variability in exosome content and source between the different studies, the consistencies that persist cannot be ignored. The first and most important aspect to take note of is that all MSC-based exosomes in this review led to the consistent or improved healing of traumatic osteochondral defects, regardless of their source. Many of these sources stimulate expression of anti-inflammatory genes while also decreasing expression of pro-inflammatory genes. Furthermore, within each type of exosome derived source, similarities in the changes of specific genes expressed can be noted. However, it is important to consider that each group did not mention all genes that experienced changes in expression. This suggests that there could be more similarities in expression than this review notes. Some similarities can be seen in the reported changes from the bone marrow MSCs through the consistent upregulation of the COL2A1 gene, increased deposition of Aggrecan, and increased presence of M2 macrophages. These changes decrease inflammation and promote healing. Bone marrow MSC-based exosomes also show a consistent downregulation of COL1A1, MMP13, and presence of M1 macrophages. These changes lead to less scarring, increases in collagen, and decreased inflammation, respectively, all of which promote healing. Although the other exosomes used may not have the same changes in gene expression, they do lead to similar overall changes at the site of the defect. These changes include an increase in anti-inflammatory factors and a decrease in pro-inflammatory factors. This localized differential expression allows for the defect to heal in a fashion that mimics the compositions of the native cartilage in the affected area compared to the negative control. This is seen by the increased hyaline cartilage with mature chondrocytes, as well as having a similar Masson and H&E staining to the positive control. 

Optimal performance of exosome–scaffold therapeutics in vivo is highly dictated by their biomechanical properties. This is especially important when considering scaffolds that are surgically implanted within cartilage defects. The ideal scaffold would be able to maintain form and size while withstanding the compressive forces of the knee joint [48]. Furthermore, newly regenerated articular cartilage from the scaffold should mimic the biomechanical behaviors of native tissue. The studies in our review examined biomechanics from these two perspectives to describe the characteristics and performance of exosome–scaffold therapies.

The inclusion of cross-linking materials such as nanoclay or alginate within viscoelastic scaffolds such as hydrogels led to improved ultimate strength and compressive moduli when compared to scaffolds alone. This can be attributed to enhancements in electrostatic interactions and covalent bonding reducing porosity and strengthening connectivity [49]. Furthermore, the addition of MSC-based exosomes to these scaffolds led to no significant changes in scaffold compressive stability or performance. In fact, Pang et al. found that inclusion of MSC-based exosomes led to decreased contraction of hydrogel molecular chains, thus improving compressive resistance [19]. It is important to note that viscoelastic scaffolds display a high degree of water absorption, leading to swelling that can compromise form and function [50,51]. The studies in our review demonstrate that the inclusion of cross-linking materials as well as exosomes helps maintain structural stability without significant swelling. Additionally, Zhang et al., in 2021, showed that their hydrogel scaffold was able to display similar adhesion and shear strengths to standard cartilage tissue adhesive, proving that the fluid environment of the knee joint would not displace or impair scaffolds following implantation [24].

Young’s modulus and stiffness were calculated for newly generated articular cartilage at various timepoints following exosome–scaffold implantation, with values ranging from 5.42 to 15.14 MPa and 2.84 to 7.94 N/mm, respectively. These values were obtained at various locations within the defect; however, we decided to focus on the overall periphery of the defect, as previous biomechanical studies have shown that the greatest concentration of compressive forces lie along the periphery of a defect region [52,53]. Across all timepoints, cartilage tissue produced by exosome–scaffold therapeutics was shown to biomechanically outperform tissue produced by treatment with either exosomes or scaffold alone. Collagen fibers produced by exosome–scaffold implants displayed organized growth and increased protein deposition, which contributed to improved stiffness and load-bearing properties. Furthermore, newly regenerated articular cartilage in exosome–scaffold therapies was shown to have no significant differences in biomechanics when compared to native tissue by as soon as 3 months post-op. This can be attributed to the induction of hyaline cartilage formation over fibrocartilage; the former is seen in native tissue, while the latter is formed within scar tissue following cartilage injury [54]. Taken together, the combination of exosomes and cross-linked scaffolds promote improved tissue morphology as well as functional mechanical support to heal defects in articular cartilage induced by traumatic OA.

This current review is not without limitations. First, there is inherent variability within the study design, interventions, and animal models across the 13 included studies. Furthermore, reported outcomes were inconsistent from paper to paper, with only a handful examining biomechanical performance or exosome release patterns. Performing a risk of bias assessment helped quantify the level of risk involved within our review, and the results show mostly low-to-unsure risk. Additionally, the sample size of studies reporting on specific outcomes was always greater than give, allowing for reasonable comparisons and conclusions to be made. There are limitations attached to the study’s search strategy. It is possible that some relevant studies were excluded due to the omission of certain terms in our preliminary search. Lastly, only studies published from 2018 to 2023 were included in our analysis. As a result, any earlier developments or trends in exosome–scaffold therapeutics were not accounted for in this study.

## 4. Materials and Methods

An organized literature search was performed using the PubMed and Embase databases. Notable studies were identified based on the following key words: “exosome”, “osteoarthritis”, “cartilage”, “chondro”, “scaffold”, and “biomaterial”. Selected citations were uploaded into Covidence (Cochrane, London, UK). The reference lists of these studies were also sorted through, with relevant papers selected for screening. Studies were included if they met the following criteria: published from 2018 to 2023, utilized in vivo animal models, induced knee OA or osteochondral defects through traumatic means (ex. ACL transection, creating lesions in femoral condyle, etc.), examined exosomes derived from any source, and utilized bioengineered scaffolds as an exosome delivery vehicle. Any non-English papers, reviews, meta-analyses, studies that analyzed in vitro or human models, studies that induced knee OA through collagenase injection, and studies that examined non-knee OA models were excluded from review. Abstract screening, full-text review, and data extraction were then conducted in accordance with PRISMA guidelines. The PRISMA flowchart seen in Figure 1 outlines the systematic review process. The studies collected and included for analysis were then reviewed for risk of bias and study design using the Cochrane risk of bias assessment tool [55].

## 5. Conclusions

The studies in this review outline the successful implementation of scaffolds as a drug delivery vehicle for MSC-based exosomes for the treatment of post-traumatic osteoarthritis of the knee (PTOK). Different scaffold materials and designs allow for control over exosome release and retention while providing biomechanical support. Furthermore, scaffolds support exosomes from a variety of different organic sources and contribute to genetic and physiologic changes that ultimately lead to the generation of new articular cartilage in the knee joint. Future studies should aim to examine the long-term effects of exosome–scaffold therapeutics in in vivo animal models to determine overall immunogenicity and biocompatibility. Furthermore, investigations into new materials and designs are also encouraged in order to further fine-tune drug delivery and tissue integration.

## Figures and Tables

**Figure 1 ijms-24-15178-f001:**
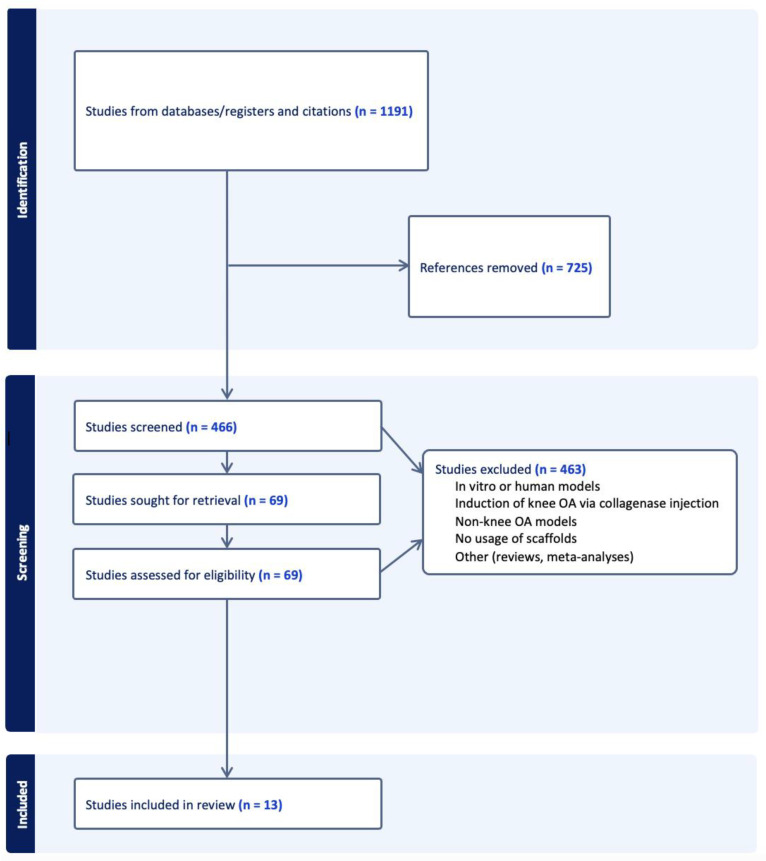
PRISMA flow diagram of the literature search conducted including criteria and selection (n = number of studies).

**Table 1 ijms-24-15178-t001:** Experimental design of selected exosome–scaffold therapy studies.

Author	Exosome Source	Exosome Dosage	Animal	Cartilage Injury/Osteoarthritis Model
Chen et al. [14].	Bone marrow MSCs	200 μg/mL	Rabbit	Drilled cylindrical defect in patellar groove
Hu et al. [15].	Human umbilical cord MSCs	44 μg/mL	Rat	Drilled cylindrical defects in articular cartilage
Jiang et al. [16].	Human Wharton’s jelly MSCs	125 μg/mL (25 μg/mL × 5 injections)	Rabbit	Drilled osteochondral defect in femoral trochlea
Liu et al. [17].	Immortalized clonal MSCs	44 μg/mL	Rabbit	Drilled cylindrical osteochondral defect in patellar groove
Liu et al. [18].	Bone marrow MSCs	50 μg/mL, 100 μg/mL	Rat	Drilled chondral defect in patellar sulcus
Pang et al. [19].	Bone marrow MSCs	80 μg/mL	Mouse	Surgical destabilization of medial meniscus
Shen et al. [20].	Bone marrow MSCs	44 μg/mL	Rat	Drilled osteochondral defects in center of patellar groove
Tao et al. [21].	Synovial MSCs	100 μg/mL	Rat	Transection of ACL, MCL, medial meniscus
Wong et al. [22].	Immortalized clonal MSCs	600 μg/mL (200 μg/mL × 3 injections)	Rabbit	Bilateral drilled osteochondral defects in femoral trochlear grooves
Yan et al. [23].	Human umbilical cord MSCs	25 μg/mL	Rat	Drilled osteochondral defect in femoral trochlea
Zhang et al. [24].	Bone marrow MSCs	200 μg/mL	Rat	Drilled osteochondral defect in femoral trochlear groove
Zhang et al. [25].	Immortalized clonal MSCs	3000 μg/mL (1000 μg/mL × 3 injections)	Micropig	Drilled osteochondral defects in medial femoral condyles

**Table 2 ijms-24-15178-t002:** Therapeutic effects of combined exosome–scaffold treatments in in vivo animal models.

Author	Scaffold Type	Exosome Release	Gross Findings	ICRS Score	Biomechanical Testing	Histology	Gene Expression
Chen et al. [14].	Hydrogel	7 days	Smooth neo-cartilage and enhanced defect filling at 6 and 12 weeks post-op	Histological: 16.1 ± 0.83 at 12 weeks post-op	N/A	Fibrocartilage + hyaline-like cartilage at 6 and 12 weeks post-op	Upregulation of M2 macrophages, COL2A1; Downregulation of M1 macrophages, MMP13
Hu et al. [15].	Hydrogel	N/A	Intact, smooth regenerated neo-tissue with complete integration with surrounding cartilage at 12 weeks post-op	Macroscopic: 10.7 ± 1.5 Histological: 11.2 ± 0.7 at 12 weeks post-op	Ultimate strength: 48.2 ± 8.1 kPa to 259.8 ± 35.6 kPa with nanoclay addition	Intense staining similar to native cartilage	Increased COL2, P-AKT, miR-23a-3p; Reduced PTEN
Jiang et al. [16].	Acellular extracellular matrix	N/A	Regenerated tissue at depth matching surrounding cartilage, somewhat visible boundary at 3 months post-op; smooth surface, no visible boundary at 6 months post-op	Highest ICRS score when compared to other groups at 3, 6 months post-op	No significant difference in Young’s modulus between regenerated and native tissue at 6 months post-op	Gap between repaired tissue and normal cartilage at 3 months post-op; completely fused with normal cartilage at 6 months post-op	Increased M2 macrophage activity; significant increase in IL-10
Liu et al. [17].	Hyaluronic Acid	14 days	Regenerated tissue with smooth surface, complete defect filling, integration with surrounding cartilage at 12 weeks post-op	Highest ICRS score when compared to other groups at 12 weeks post-op	N/A	Newly formed tissue was mainly hyaline cartilage	COL2 upregulation; COL1 downregulation
Liu et al. [18].	Hydrogel	N/A	The articular surface was smoother with integration with adjacent host cartilage at 8 weeks pot-op	Highest ICRS score when compared to other groups at 4, 6, and 8 weeks post-op	N/A	Regular presence of mature chondrocyte cells indicative of predominate hyaline cartilage formation	COL2A1, Agg, Prg4, SOX-9 upregulation Col1A1, Adamts5, IL-1b, C-myc downregulation
Pang et al. [19].	Hydrogel	30 days with 100% release	Smooth surface in repaired cartilage at 4 weeks post-op	OARSI score -> significant reduction in cartilage lesion severity when compared to other groups at 4 weeks post-op	Compressive strength: 16.779 kPa (compared to 12.096 kPA in scaffold alone) High cyclic compression stability	Increased polysaccharide content in repaired cartilage at 4 weeks post-op	COL2, Agg, M2 macrophage upregulation MMP13, TNF-a, IL-1B, M1 macrophage downregulation
Shen et al. [20].	Hydrogel	30 days with 85–89% release	Neo-tissues in region adjacent to defects were smooth and complete with fusion with surrounding normal cartilage at 12 weeks post-op	Macroscopic: 11.25 ± 0.96 at 12 weeks post-op	N/A	Smooth, flat surfaces with regenerated articular chondrocytes arranged regularly	COL2 upregulation; COL1 downregulation
Tao et al. [21].	Hydrogel	35 days with 80% release	N/A	N/A	N/A	Reversal of OA cartilage damage, protective ECM	N/A
Wong et al. [22].	Hyaluronic Acid	N/A	Greater neo-tissue filling, smooth surface regularity at 12 weeks post-op	Macroscopic: 8.75 ± 0.87 at 6 weeks post-op; Macroscopic: 10.33 ± 0.49 at 12 weeks post-op	Young’s Modulus: 7.34 ± 0.67 MPa at 6 weeks post-op 12.22 ± 3.67 MPa at 12 weeks post-op Stiffness: 3.85 ± 0.48 N/mm at 6 weeks post-op 6.40 ± 1.92 N/mm at 12 weeks post-op	Thicker than normal cartilage with high cellularity >40% hyaline cartilage at 6 weeks post-op; >80% hyaline cartilage at 12 weeks post-op	Greater areal deposition of GAG; Lower areal deposition of COL1
Yan et al. [23].	Acellular extracellular matrix	5 days (via fluorescence)	Mostly filled cartilage defect area with uneven surface and slight border with surrounding tissue at 4 weeks post-op; new tissue was smooth with no obvious border with surrounding tissue at 8 weeks post-op	Highest ICRS score when compared to other groups at 4 and 8 weeks post-op	N/A	Small amount of fibrous tissue with increased chondrocytes at 4 weeks post-op; high level of chondrocytes with normal arrangement at 8 weeks post-op	Abundant expression of proteoglycans and COL II at 4 and 8 weeks post-op
Zhang et al. [24].	Hydrogel	14 days with 87% release	Smooth, continuous cartilage surface with rarely distinct boundary with surrounding tissue at 6 and 12 weeks post-op	Highest ICRS score when compared to other groups at 6 and 12 weeks post-op	Adhesive strength: 121.7 ± 12.3 kPa	Newly formed cartilage in the defect at 6 weeks post-op; smooth surface similar to the normal at 12 weeks post-op	COL2, Agg upregulation; COL1 downregulation
Zhang et al. [25].	Hyaluronic Acid	N/A	Significantly improved degree of defect repair, integration to border zone, macroscopic appearance	Macroscopic: 9.22 ± 1.94 at 4 months post-op Histological: 79.71 ± 12.3 at 4 months post-op	Young’s Modulus: 19.58 ± 6.79 MPa at 4 months post-op; Stiffness: 10.25 ± 3.56 N/mm at 4 months post-op	Newly formed cartilage with matrix staining and tissue and cell morphologies resembling adjacent native cartilage at 4 months post-op	N/A

**Table 3 ijms-24-15178-t003:** Risk of bias assessment using Cochrane risk of bias assessment tool.

Author	Random Sequence Allocation (Selection Bias)	Allocation Concealment (Selection Bias)	Blinding of Personnel (Performance Bias)	Blinding of Outcome Assessment (Detection Bias)	Incomplete Outcome Data (Attrition Bias)	Selective Reporting (Reporting Bias)	Other Sources of Bias
Chen et al. [14].	Unsure risk	Unsure risk	Unsure risk	Low risk	Low risk	Low risk	Low risk
Hu et al. [15].	Low risk	Unsure risk	Unsure risk	Low risk	Low risk	Low risk	Low risk
Jiang et al. [16].	Low risk	Unsure risk	Low risk	Low risk	Low risk	Low risk	Low risk
Liu et al. [17].	Low risk	Unsure risk	Unsure risk	Low risk	Low risk	Low risk	Low risk
Liu et al. [18].	Low risk	Unsure risk	Low risk	Low risk	Low risk	Low risk	Low risk
Pang et al. [19].	Low risk	Unsure risk	Unsure risk	Low risk	Low risk	Low risk	Low risk
Shen et al. [20].	Unsure risk	Unsure risk	Unsure risk	Low risk	Low risk	Low risk	Low risk
Tao et al. [21].	Unsure risk	Unsure risk	Unsure risk	Low risk	Low risk	Low risk	Low risk
Wong et al. [22].	Low risk	Unsure risk	Low risk	Low risk	Low risk	Low risk	Low risk
Yan et al. [23].	Low risk	Unsure risk	Low risk	Low risk	Low risk	Low risk	Low risk
Zhang et al. [24].	Low risk	Unsure risk	Unsure risk	Low risk	Low risk	Low risk	Low risk
Zhang et al. [25].	Low risk	Unsure risk	Low risk	Low risk	Low risk	Low risk	Low risk

## Data Availability

Not applicable.

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
