# Peer review of "Exosome-Laden Scaffolds for Treatment of Post-Traumatic Cartilage Injury and Osteoarthritis of the Knee: A Systematic Review"

_ijms, 2023, doi:10.3390/ijms242015178_

Round 1

Reviewer 1 Report

I think it is a well-written review.

Author Response

We would like to express our gratitude for the reviewer’s comments and feedback on our manuscript. We received suggestions for edits and revisions from other peer review feedback; as a result, we would be appreciative if the reviewer read over our revised manuscript and provided any necessary comments, insights, or suggestions.

Reviewer 2 Report

This is a very interesting manuscript about the preclinical evidence of EV-laden scaffolds in cartilage injury and osteoarthritis. MSC-EVs have a high immunomodulative and regenerative potential and are therefore a recent research focus in this field. The manuscript is well written. However, the following aspects need to be addressed.

1) Title: The authors defined all used models as posttraumatic OA models. This is not correct. Osteochondral defect models are not an OA model, but an injury model, whereas ACL or meniscus resection are well known OA models. Please revise the title.

2) In the title the authors use the term “EV”, whereas in the manuscript they use the term “exosome”. Please provide a definition of EVs and the different subtypes of EVs and change the title if only exosomes were analyzed

3) Line 78: Please provide also the number of particles/ EVs measured by NTA

4) Results: Please also report from each study if the MISEV guidelines were followed (Report of NTA, Electron microscopy, Western blot)

5) Lines 105 ff. Please provide references

6) Lines 116 ff. Please report from which region the acellular extracellular matrices were taken

7) Line 200: Please write bone marrow instead of bone (revise in the whole manuscript)

8) Discussion: Discuss the potential mechanism behind the exosome effect and the advantage of scaffolds in more detail.

Author Response

  1. Title: The authors defined all used models as posttraumatic OA models. This is not correct. Osteochondral defect models are not an OA model, but an injury model, whereas ACL or meniscus resection are well known OA models. Please revise the title.

We agree with the reviewer’s comments and have adjusted our review title accordingly (Lines 2-4). We further wanted to clarify that trauma-induced osteochondral defects can lead to eventual post-traumatic osteoarthritis. Lines 33-35 and Line 292 have been modified to account for this clarification and a citation was added to support our definition (Lines 330-332).

  1. In the title the authors use the term “EV” whereas in the manuscript they use the term “exosome”. Please provide a definition of EVs and the different subtypes of EVs and change the title if only exosomes were analyzed.

We agree with the reviewer’s comments. Lines 49-53 have been modified to provide a definition of EVs as well as the two major subclasses: exosomes and microvesicles. Furthermore, we highlighted the reason why exosomes are considered as the primary effector molecule amongst EVs when compared to microvesicles. Citations were added as needed (Lines 345-346). Since our review looks exclusively at exosomes, we have modified our title accordingly (Lines 2-4).

  1. Line 78: Please provide also the number of particles/EVs measured by NTA.
  2. Results: Please report from each study if the MISEV guidelines were followed (Report of NTA, Electron microscopy, Western blot).

As per the reviewer’s suggestion, we have provided a dosage range in particles/EV as reported by the studies (Lines 81-82). We would like to clarify that we have maintained our units of exosome dosage in Table 1 as micrograms/mL as a means of standardization, as not all papers reported exosome dosage in particles/EV. 2 dosage values in Table 1 were revised and are highlighted in the revised manuscript (Line 76). Lastly, a sentence was added to verify that all studies adhered to MISEV guidelines (Lines 84-85). A citation was added for reference (Lines 396-400).

  1. Lines 105 ff. Please provide references.

We agree with the reviewer’s comment and have added references for all studies in Table 1 (Lines 355-395). We would like to note that the study by Cheng et al. published in 2022 was removed from our manuscript, as it was recently retracted from publication. Adjustments were made throughout the revised manuscript as necessary to account for this removal.

  1. Lines 116 ff. Please report from which region the acellular extracellular matrices were taken.

We would like to thank the reviewer for pointing out this point of clarification. Lines 120-121 highlight that the acellular extracellular matrices were derived from porcine knee joint articular cartilage.

  1. Line 200: Please write bone marrow instead of bone (revise in whole manuscript).

We agree with the reviewer’s suggestion and have made changes accordingly.

  1. Discussion: Discuss the potential mechanism behind the exosome effect and the advantage of scaffolds in more detail.

We thank the reviewer for the suggestion, as it provides further details that highlight the importance and relevance of our review. Lines 168-174 highlight the postulated mechanisms of actions of exosomes in treating knee OA. Lines 187-198 describe the advantages that scaffolds provide in creating an environment conducive to improved exosome activity and controlled release into the environment. Relevant references were added (Lines 447-449, 475-477).

Reviewer 3 Report

The authors review mostly studies that have used exosomes in scaffolds for cartilage repair in animals. Two of the studies in the review used exosomes in scaffolds for the treatment of PTOA. Therefore, the title of this study is misleading, because most of the studies reviewed in this manuscript analyzed the degree of cartilage repair and not the degree of PTOA. In addition, it is not mentioned whether the exosome containing scaffolds were injected and/or placed in the defect. If it is injected the scaffold is not expected to have the mechanical properties of cartilage. If, however, it is used to placed in the defect then the scaffold should have similar mechanical properties as cartilage and be biodegradable over time. However, the authors do not discuss any of these important issues. In addition, no clear mention from which cells these exosomes were isolated in the different studies, and more importantly how these exosomes were isolated. Finally, none of the studies which were reviewed in this manuscript are being cited in the reference section. In summary, this review is not very informative and poorly written.

Minor:  PTOK is a strange abbreviation for post-traumatic osteoarthritis.

Author Response

The authors review mostly studies that have used exosomes in scaffolds for cartilage repair in animals. Two of the studies in the review used exosomes in scaffolds for the treatment of PTOA. Therefore, the title of this study is misleading, because most of the studies reviewed in this manuscript analyzed the degree of cartilage repair and not the degree of PTOA.

We appreciate the reviewer’s comments here and acknowledge the discrepancy between the animal models chosen for the study and our intended focus. We would like to highlight that osteochondral defects have been shown to lead to the development of post-traumatic osteoarthritis over time. As a result, our review aims to examine animal models of both osteochondral defects along with traditional post-traumatic OA models such as ACL and menisci resections. We have modified our title as per the reviewer’s suggestion (Lines 2-4) and have made adjustments throughout the manuscript to indicate our reasoning for analyzing both cartilage injury and knee OA animal models.

In addition, it is not mentioned whether the exosome containing scaffolds were injected and/or placed in the defect. If it is injected the scaffold is not expected to have the mechanical properties of cartilage. If, however, it is used to placed in the defect then the scaffold should have similar mechanical properties as cartilage and be biodegradable over time. However, the authors do not discuss any of these important issues.

We agree with the reviewer’s assessment and have added details regarding the placement of scaffolds within animal knee models (Lines 114-115, 125-127, 136-138). Furthermore, Lines 253-254 highlight the importance of biomechanical considerations for scaffolds specifically implanted within cartilage defects. We would like to note that our discussion of biomechanics chose to focus on survivability within an in vivo environment as well as characteristics of regenerated tissue as these are key issues that were noted across our selected studies. The need for further innovation and design of exosome-scaffold therapeutics stems from a need to optimize scaffold properties in response to biophysical cues and improve the quality of articular cartilage that is generated via exosome action.

In addition, no clear mention from which cells these exosomes were isolated in the different studies, and more importantly how these exosomes were isolated.

Our study focuses on exosomes derived from mesenchymal stem cells harvested from different locations. Table 1 the different MSC sources, ranging from bone marrow to human Wharton’s jelly to immortalized clonal lines. We agree with the reviewer’s comment regarding detailing of the exosome isolation process and have made revisions to accommodate for this (Lines 87-89).

Finally, none of the studies which were reviewed in this manuscript are being cited in the reference section.

We agree with the reviewer’s comment and have added references for all studies in Table 1 (Lines 355-395). We would like to note that the study by Cheng et al. published in 2022 was removed from our manuscript, as it was recently retracted from publication. Adjustments were made throughout the revised manuscript as necessary to account for this removal.

We thank you again for giving us the opportunity to respond to the reviewers’ comments and upload a revised manuscript. We strongly hope that you reconsider our updated systematic review for publication, as we believe it can help guide further research and innovation to develop biologic solutions to treat cartilage injury and post-traumatic osteoarthritis.

Round 2

Reviewer 2 Report

Thanks to the authors. All comments were adressed in detail.

Author Response

(The authors gave the same response as above.)

Reviewer 3 Report

The authors have adequately addressed most of the concerns raised by these reviewer.

The authors, however, have added a new paragraph (Pages 6-7, lines 196-211). Especially, the statement "An in vitro study by Rocha et. al in 2018 showed 200 that when cultured in three-dimensional scaffolds, exosomes demonstrate an upregulation of microRNA expression into the culture environment" needs to be clarified, since exosomes cannot generate new RNA.   

Author Response

We appreciate the reviewer’s comments here and acknowledge the confusion that stems from our phrasing of exosome behavior within three-dimensional culture environments. We have modified our description of the results of the study by Rocha et. al in 2018 to clarify that 3D environments favor upregulation of certain microRNA lines that thereby enhance exosome activity (Lines 200 - 203).
